# Reconstructing long-term (1980-2022) daily ground particulate matter concentrations in India (LongPMInd)

Shuai Wang[1], Mengyuan Zhang[1], Hui Zhao[2], Peng Wang[3,4], Sri Harsha Kota[5], Qingyan Fu[6], Cong Liu[7], Hongliang Zhang[1,4,8*]

[1]Department of Environmental Science and Engineering, Fudan University, Shanghai 200438, China
[2]School of Resources and Environmental Engineering, Jiangsu University of Technology, Changzhou 213001, China
[3]Department of Atmospheric and Oceanic Sciences, and Institute of Atmospheric Sciences, Fudan University, Shanghai, 200438, China
[4]IRDR ICoE on Risk Interconnectivity and Governance on Weather/Climate Extremes Impact and Public Health, Fudan University, Shanghai, China
[5]Department of Civil Engineering, Indian Institute of Technology, Delhi, 110016, India
[6] Shanghai Academy of Environmental Sciences, Shanghai 200003, China
[7]School of Public Health, Fudan University, Shanghai, 200032, China
[8]Institute of Eco-Chongming (IEC), Shanghai 200062, China

*Correspondence to*: Hongliang Zhang (zhanghl@fudan.edu.cn)

**Abstract.** Severe airborne particulate matter (PM, including $PM_{2.5}$ and $PM_{10}$) pollution in India has caused widespread concern. Accurate PM concentrations are fundamental for scientific policymaking and health impact assessment, while surface observations in India are limited due to scarce sites and uneven distribution. In this work, a simple structured, efficient, and robust model based on the Light Gradient Boosting Machine (LightGBM) was developed to fuse multi-source data and estimate long-term (1980-2022) historical daily ground PM concentrations in India (LongPMInd). The LightGBM model shows good accuracy with out-of-sample, out-of-site, and out-of-year cross-validation CV test $R^2$ of 0.77, 0.70, and 0.66, respectively. Small performance gaps between $PM_{2.5}$ training and testing (delta RMSE of 1.06, 3.83, and 7.74 µg m$^{-3}$) indicate low overfitting risks. With great generalization ability, the open-accessible, long-term, and high-quality daily $PM_{2.5}$ and $PM_{10}$ products were then reconstructed (10 km, 1980-2022). It shows that India has experienced severe PM pollution in the Indo-Gangetic Plain (IGP), especially in winter. PM concentrations significantly increased (p<0.05) in most regions since 2000 (0.34 µg m$^{-3}$ year$^{-1}$). The turning point occurred in 2018 when the Indian government launched the National Clean Air Program, $PM_{2.5}$ concentrations declined in most regions (-0.78 µg m$^{-3}$ year$^{-1}$) during 2018-2022. Severe $PM_{2.5}$ pollution caused continuous increased attributable premature mortalities, from 0.73 (95 % CI: 0.65-0.80) million in 2000 to 1.22 (95 % CI: 1.03-1.41) million in 2019, particularly in the IGP, where attributable mortality increased from 0.36 to 0.60 million. The LongPMInd has the potential to support multi-applications of air quality management, public health, and climate change. The daily and monthly $PM_{2.5}$ and $PM_{10}$ concentrations are publicly accessible at https://doi.org/10.5281/zenodo.10073944 (Wang et al., 2023a).

# 1 Introduction

Airborne particulate matter (PM, including $PM_{2.5}$ with diameters < 2.5 μm and $PM_{10}$ with diameters < 10 μm) not only impacts
climate by changing radiation budgets but also has significant adverse effects on human health(Murray et al., 2020; Wang et al., 2012; Yang et al., 2016). India is one of the most populous countries, with severe PM pollution resulting from rapid economic development and industrialization over the last few decades. Exposure to $PM_{2.5}$ has become one of the leading causes of health burden in India, including heart disease, stroke, lung cancer, and premature death (Pandey et al., 2021; Dandona et al., 2017).

Accurate ground PM concentrations are prerequisites for evidence-based policymaking and health impact assessments. The Central Pollution Control Board (CPCB) of India has established and maintained ground-based monitoring networks with ~335 continuous ambient air quality monitoring stations (CAAQMS) currently. However, these monitoring sites are unevenly distributed (mainly located in urban, residential, and industrial areas), with limited number of sites (monitoring density: ~0.6 sites per million population) (Brauer et al., 2019), and many cities even have no monitoring sites (Martin et al., 2019). Therefore, the surface observations alone are not sufficient to support air quality management, especially on a regional scale (Pant et al., 2019; Dey et al., 2020).

Two main approaches have been used for large-scale and long-term $PM_{2.5}$ estimation: scaling methods and statistical methods. Scaling methods use chemical transport modeling (CTM) to simulate the association between aerosol optical depth (AOD) and $PM_2$, which requires no ground observations. However, the relationship between $PM_{2.5}$-AOD is spatially and temporally variable and without the constraints of ground observations, this method usually has a large uncertainty (Ma et al., 2022). Compared with scaling methods, statistical methods based on multivariate data fusion have higher prediction accuracy and have been widely used. Statistical models (traditional linear and nonlinear regression and machine learning algorithms) estimate $PM_{2.5}$ concentrations by fitting the relationship between input variables (meteorological, emissions, and other relevant variables) and target variables (Wang et al., 2023d; Wei et al., 2021a; Ren et al., 2022b; Katoch et al., 2023).

Tree-based machine learning (ML) models typically outperform deep learning approaches and traditional machine learning methods in tabular data (e.g., air pollutant observation datasets), and thus have been widely developed and used (Grinsztajn et al., 2022; Sayeed et al., 2022). Wei et al. (2021a) and Li et al. (2021) reconstructed long-term $PM_{2.5}$ data records in China by fusing satellite, meteorological, and emission data using a spatiotemporal tree-based model. Ni et al. (2024) analyzed the contribution of meteorology and emissions to $O_3$ in China using the chemical transport model (GEOS-Chem) and a tree-based model. Sayeed et al. (2022) improved the $PM_{2.5}$ concentration in the continental United States using the Random Forest approach coped with meteorology and aerosol species of MERRA-2.

Some studies have estimated $PM_{2.5}$ concentrations in India using different methods. Chowdhury et al. (2019) used the $PM_{2.5}$–AOD equation method to estimate $PM_{2.5}$ concentrations in Delhi, however, AOD satellite data suffers from significant non-random misses, especially during cloud cover and hazy polluted days, so it is difficult to derive a spatiotemporal full-coverage PM dataset (Wang et al., 2023d; Bai et al., 2022). Bali et al. (2021) and Dey et al. (2020) estimated total $PM_{2.5}$ in India through

empirical coefficients and the MERRA2 dataset, while these coefficients vary with geographic location and pollution scenarios, which makes the estimation potentially unreliable. Kumar et al. (2023) analyzed $PM_{2.5}$ for India using a Random Forest model, which shows a gap between train and test scores with the risk of overfitting. In addition, global-scale daily $PM_{2.5}$ concentration datasets (including India) have recently been developed, including GlobalHighAirPollutants (GHAP) (Wei et al., 2023), The

70 Long-term Gap-free High-resolution Air Pollutants concentration dataset version 2 (LGHAPv2) (Bai et al., 2024). Global monthly $PM_{2.5}$ datasets have also been developed before (Van Donkelaar et al., 2021). These datasets were trained on a global scale and estimated $PM_{2.5}$ concentrations for the India region. The severity of $PM_{2.5}$ pollution in India is much higher than in Europe and the United States (Wei et al., 2023). However, due to the small number of observations recorded, the global model can learn limited knowledge of $PM_{2.5}$ pollution in India during the training process. So, the reliability and robustness of global

model estimates of $PM_{2.5}$ concentrations in India should be systematically assessed. Building a model locally in India can be a useful comparison method, which can also complement the biases in global modeling (e.g., focusing more on lightly polluted regions such as Europe and the United States). However, it is challenging to establish long-term, full-coverage, high-accuracy, open-source PM data products locally in India due to insufficient observational data and lack of model robustness due to variations of data distribution across regions and years (Kumar et al., 2023; Dey et al., 2020).

To improve performance, previous models usually have high complexity, such as numerous trees and leaf nodes (Zhang et al., 2021; Huang et al., 2021). This practice raises the requirement of computational resources and is prone to overfitting, leading to a large gap between the performance of the training and testing(Zhang et al., 2021; Jabbar and Khan, 2015; Ying, 2019). Therefore, it is necessary to minimize model complexity to avoid overfitting. The Light Gradient Boosting Machine (LightGBM) is an optimized Gradient Boosting Decision Tree (GBDT) (Ke et al., 2017a), and has shown superior performance

in many fields (Wei et al., 2021b; Yan et al., 2021; Sun et al., 2020; Liang et al., 2020). LightGBM uses Histogram's decision tree algorithm along with Gradient-based One-Side Sampling (GOSS), which can save memory and computation time (Ke et al., 2017a). Our previous study comparing several commonly used machine learning models found that the LightGBM has similar performance to the eXtreme Gradient Boosting (XGBoost) with the highest accuracy, but LightGBM was faster and more robust, which has the potential to estimate long-term concentrations of PM in India (Wang et al., 2023b).

In this work, a simple structured, efficient, and robust model based on LightGBM was developed to estimate PM concentration. Three cross-validation methods and separate test datasets were designed to evaluate model performance. Long-term (1980-2022) and open-source datasets with a spatial resolution of 10 km of $PM_{2.5}$ and $PM_{10}$ in India were then generated, and the mortalities due to $PM_{2.5}$-induced diseases were also estimated. The concentration datasets could help with pollution formation analysis, assessment of PM health risks, and air quality management in India.

## 2 Materials and methods

### 2.1 Data sources

Table 1 shows the multisource datasets used in this study. Ground observations of $PM_{2.5}$ and $PM_{10}$ during 2018-2022 in India were collected from the CPCB air quality monitoring network (www.cpcb.nic.in). The location of monitoring sites is shown in Fig. S1. As extreme values affect model robustness, Observations data less than 0.01 % and larger than 99.99 % were excluded. The fifth generation ECMWF atmospheric reanalysis datasets ERA5-Land in 1980-2022 were collected. The feature was selected by the relative importance, which was calculated by the Gain, and several meteorological factors with high relative importance are included (Table 1). Datasets of Modern-Era Retrospective analysis for Research and Applications, Version 2 (MERRA-2) in 1980-2022 were also collected, including aerosol optical depth and aerosol components and precursors (black carbon, organic carbon, sulfate, dust, and $SO_2$).

**Table 1: Summary of the ERA5, MERRA2, and ground observation data used in this study.**

| Type | Variable | Description | Spatial Resolution | Temporal Resolution |
|---|---|---|---|---|
| ERA5 | SSRD | Surface solar radiation | 0.1° × 0.1° | Hourly |
| | BLH | Boundary layer height | 0.25°× 0.25° | Hourly |
| | EVAP | Evaporation | 0.1°× 0.1° | Hourly |
| | TEMP2 | 2m air temperature | 0.1°× 0.1° | Hourly |
| | DEWP2 | 2m dewpoint temperature | 0.1°× 0.1° | Hourly |
| | SP | Surface pressure | 0.1°× 0.1° | Hourly |
| | TPREC | Total precipitation | 0.1°× 0.1° | Hourly |
| | TCLOUD | Total cloud cover | 0.25°× 0.25° | Hourly |
| | UWIND10 | 10m u component of wind | 0.1°× 0.1° | Hourly |
| | VWIND10 | 10m v component of wind | 0.1°× 0.1° | Hourly |
| MERRA2 | BCSMASS | Black carbon surface mass concentration | 0.5 °× 0.625 ° | Hourly |
| | OCSMASS | Organic carbon surface mass concentration | 0.5 °× 0.625 ° | Hourly |
| | DUSMASS25 | Dust– $PM_{2.5}$ surface mass concentration | 0.5 °× 0.625 ° | Hourly |
| | DUSMASS | Dust surface mass concentration | 0.5 °× 0.625 ° | Hourly |
| | SO2SSMASS | Sulfur dioxide surface mass concentration | 0.5 °× 0.625 ° | Hourly |
| | SO4SMASS | Sulfate surface mass concentration | 0.5 °× 0.625 ° | Hourly |
| | TOTEXTTAU | Total aerosol extinction [550 nm] | 0.5 °× 0.625 ° | Hourly |
| Observation | $PM_{2.5}$, $PM_{10}$ | Particulate matter | Point | Hourly |

## 2.2 Model building

In this study, LightGBM (Ke et al., 2017b), an efficient Gradient Boosting Decision Tree (GBDT), was used to estimate $PM_{2.5}$ and $PM_{10}$, which has been proven to be accurate, fast, and robust in our previous studies (Wang et al., 2023b; Wang et al., 2023c). Grid search cross-validation (CV) method was used to select the optimal hyperparameters. An algorithm for hyperparameter selection (SI: Algorithm 1) was designed to ensure the model's generalization ability. Loop to increase the model complexity (e.g., number of trees), ending the loop and returning the hyperparameters when the model predicted RMSE does not decrease significantly ($< 0.01$) or the difference between training and predicted RMSE does not increase significantly ($< 0.05$). Features were selected based on their relative importance. Ten meteorological features, six emission-related features, and total aerosol extinction were used to train the LightGBM and estimate PM concentrations (Fig. 1). The meteorological and emission features contributed 64% and 31% to the $PM_{2.5}$ prediction.

Meteorology is more important than emissions. Compared to MERRA5, which has higher uncertainty and lower spatial resolution, ERA5 has higher resolution and accuracy, and the meteorological features can provide richer information and contribute more in model training, thus having higher importance (Muñoz-Sabater et al., 2021; Hersbach et al., 2020). Besides, more numbers of meteorological features were used to train the model, thus contributing more to prediction results. The highest importance of surface pressure can be attributed to the important effect to $PM_{2.5}$ concentration and its high data quality (Chen et al., 2020; Bauer et al., 2015).

Three independent CV methods and three metrics (coefficient of determination: $R^2$, root mean square error: RMSE, and mean absolute error: MAE) were designed to evaluate the model's spatiotemporal predictive power. The first is out-of-sample CV, where the dataset is randomly divided into 10 subsets, one of which is taken in turn for testing, and the remaining 9 subsets are used for training, which is repeated 10 times and averaged. The second is out-of-site CV, which is similar to the out-of-sample CV, but the dataset is randomly divided by site. This method can measure the model's spatial predictive power. The third method is interannual out-of-year CV, which sequentially takes one year of data for testing and the rest for training. This approach can measure the model's predictive power for the years with no observations. Besides, observations in January-June 2023 were used as a separate test set, and these data were not involved in any of the training and hyperparameter selection processes.

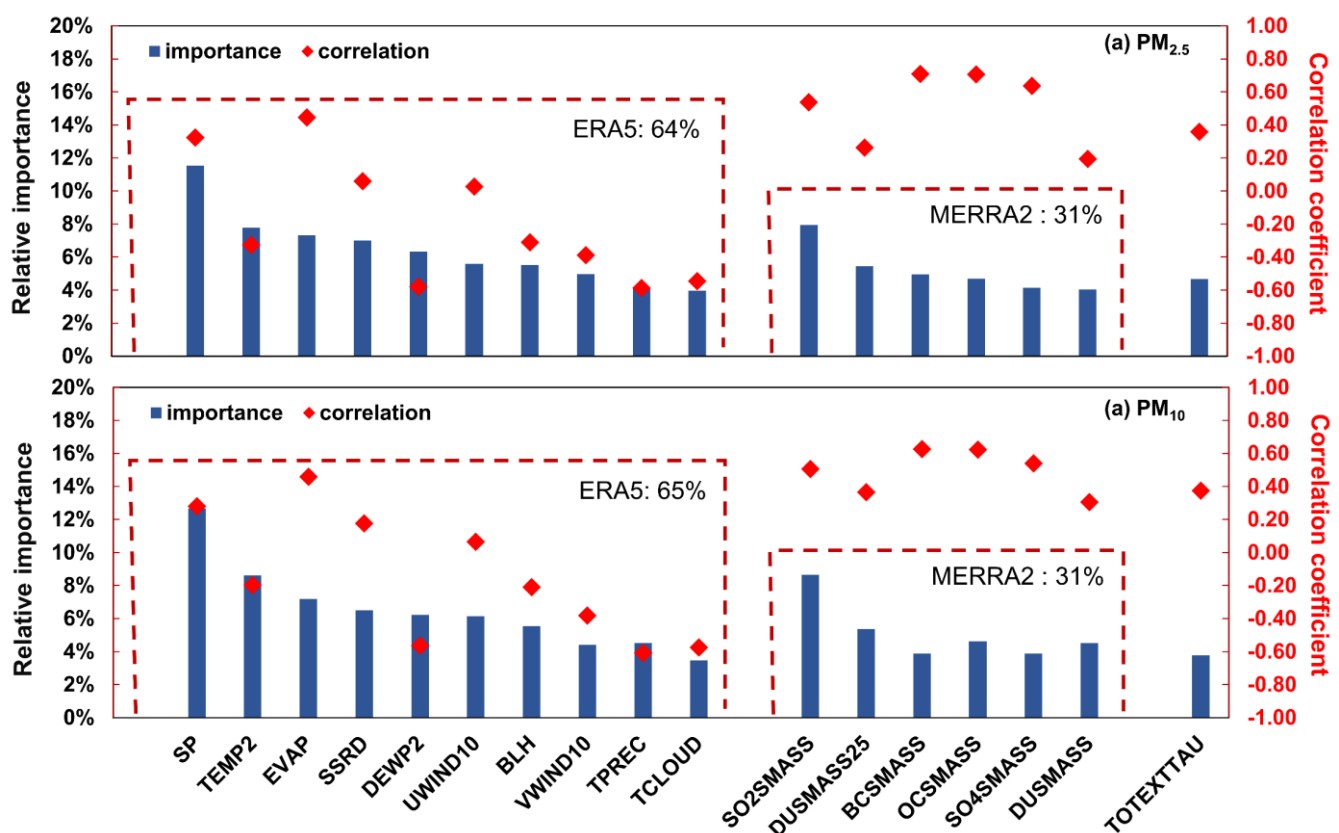

**Figure 1: Relative importance and correlation coefficient for the PM$_{2.5}$ and PM$_{10}$ estimates models. Description of the features is shown in Table 1.**

**2.3 Mortality estimation.**

According to the database of Global Burden of Disease Study (GBD) in 2019 (Murray et al., 2020; Vos et al., 2020), annual average concentrations were used to assess long-term exposure to PM$_{2.5}$, and premature deaths were assessed using the following equation:

$$M_{y,i,j} = \frac{RR_j(C_{y,i})-1}{RR_j(C_{y,i})} \times P_{y,i} \times I_{y,j} \tag{1}$$

Where, M$_{y, i}$ represents the mortality attributable to cause j due to long-term PM$_{2.5}$ exposure in year y in region i. RR(C$_{y, i}$) represents the relative risk of cause j for year y in region i. P$_{y, i}$ represents the population j in year y in region i, and I$_y$ represents the baseline mortality in year y.

PM$_{2.5}$ exposure-related deaths due to ischemic heart disease (CVD_IHD), chronic stroke (CVD_stroke), obstructive pulmonary disease (RESP_COPD), lung cancer (NEO_LUNG), lower respiratory infections (LRI), and diabetes mellitus type II (T2_DM)

were estimated. The gridded population data was obtained from the WorldPop datasets (https://www.worldpop.org). The relative risk is a discrete value obtained from GDB 2019, which is the relative risk corresponding to each PM$_{2.5}$ concentration

level. Details about the calculation way of relative risk can be found in GBD 2019 (Murray et al., 2020). Annual baseline mortality (2000-2019) and risk of cause-specific deaths at different $PM_{2.5}$ levels were obtained from GBD 2019. The minimum-risk exposure level for the health effects of $PM_{2.5}$ is in the range of 2.4 to 5.9 µg m$^{-3}$.

## 3 Results

### 3.1 Long-term India $PM_{2.5}$ dataset

Applying the trained LightGBM model to the large input dataset constructed for the years 1980 - 2022, the long-term high-quality daily $PM_{2.5}$ and $PM_{10}$ products of India (LongPMInd) are reconstructed. Table 2 summarizes the basic information about the dataset, the data is provided in NetCDF format with a spatial resolution of 10 km. LongPMInd dataset to the best of our knowledge is the first open-source, longest term (i.e. 1980-2022) and relatively high accuracy dataset covering the entire India. The daily, monthly, and yearly $PM_{2.5}$ and $PM_{10}$ datasets are publicly available at https://doi.org/10.5281/zenodo.10073944 (Wang et al., 2023a).

**Table 2: Summary of the LongPMInd dataset**

| Data description | LongPMInd dataset |
|---|---|
| Data type | Gridded |
| File format | NetCDF |
| Specie | $PM_{2.5}$, $PM_{10}$ |
| Spatial reference | WGS 84 |
| Horizontal resolution | $0.1° \times 0.1°$ ($\approx$ 10 km $\times$ 10 km) |
| Horizontal coverage | India, [60° E, 100° E], [5.0° N, 40.0° N] |
| Temporal resolution | Daily, monthly, and yearly |
| Temporal coverage | 1980-2022 |

### 3.2 Model performance

Table 3 shows the training and testing results of out-of-sample CV, out-of-site CV, and out-of-year CV for daily $PM_{2.5}$ and $PM_{10}$. Overall, the model shows good accuracy with out-of-sample CV $R^2$ of 0.77, 0.76, and RMSE of 29.57, 51.63 µg m$^{-3}$ for daily $PM_{2.5}$ and $PM_{10}$. Monthly predictions show better performance with out-of-sample CV $R^2$ of 0.87, 0.86, and RMSE of 17.65, 31.26 µg m$^{-3}$ for monthly $PM_{2.5}$ and $PM_{10}$. More importantly, out-of-sample CV results of training and testing showed small accuracy gaps with RMSE and MAE of 1.06 (4 %) and 0.51 (3 %) µg m$^{-3}$ for $PM_{2.5}$, and 1.52 (3 %) and 0.9 (3 %) µg m$^{-3}$ for $PM_{10}$ reflecting good generalization ability. Out-of-site CV measures the model's predictive ability for unobserved areas. The spatially validated $R^2$ and RMSE for $PM_{2.5}$ and $PM_{10}$ were 0.70, 0.65, and 31.73, 51.37 µg m$^{-3}$, respectively, indicating the model's ability to fill the unobserved areas accurately. The small performance gap between out-of-site CV training and

testing also reflects good spatial generalization ability. Observations before 2018 are limited due to the number and quality of sites. Out-of-year CV was used to evaluate LightGBM prediction performance, which was conducted by sequentially taking one-year data for testing and the rest for training. The model's prediction accuracy for unobserved years decreases slightly compared to out-of-sample CV ($R^2$ decreases by 14% and RMSE increases by 20%) due to differences in data distribution among years (Fig. S2). Notably, most predictions are consistent with observations, with most data samples evenly distributed around the 1:1 line (Fig. S3), but with the underestimation for high PM levels and overestimation for low PM levels (slopes:0.75 and 0.74, intercepts: 16.45 and 35.79 μg m$^{-3}$ for daily $PM_{2.5}$ and $PM_{10}$ predictions). Monthly predictions show better agreement with observations with slopes of 0.84 and 0.82, and intercepts of 10.26 and 23.53 μg m$^{-3}$ for monthly $PM_{2.5}$ and $PM_{10}$. The under- and over-estimation indicate the potential unreliability of model predictions for extreme pollution and extreme clean days. This can be attributed to the small proportion of data records for extreme pollution and clean days.

Observations from January to June in 2023 were used for testing, which were not involved in any training or hyperparameter selecting processes (Fig S4 and Table S1). Six representative regions were selected for the analysis including Delhi and Uttar Pradesh (IGP region), Gujarat (Western India region), Madhya Pradesh (Central India region), West Bengal (Eastern India region), and Andhra Pradesh (Southern India region). The model shows accurate prediction ability with RMSE of 33.58 and 64.25 μg m$^{-3}$ for $PM_{2.5}$ and $PM_{10}$ respectively in India. The model can capture the decreasing trend of PM concentration from January to June in different regions of India but with some biases, e.g., overestimation of $PM_{2.5}$ in Uttar Pradesh on 8 January; and underestimation of haze pollution in Gujarat on 19 February. The large RMSE of $PM_{2.5}$ prediction in Uttar Pradesh (32.72 μg m$^{-3}$) could be attributed to the complexity of pollution causes in the region as well as insufficient observation data. The small RMSE (8.34 μg m$^{-3}$) of $PM_{2.5}$ prediction in Andhra Pradesh can be related to the light haze pollution and small fluctuation of $PM_{2.5}$ concentration.

**Table 3: Training and testing results of out-of-sample CV, out-of-site CV, and out-of-year CV for daily $PM_{2.5}$ and $PM_{10}$ (2018-2022). RSME and MAE unit: μg m$^{-3}$.**

| Spec | Type | $R^2$ | | RMSE (μg m$^{-3}$) | | MAE (μg m$^{-3}$) | |
|------|------|------|-------|------|-------|------|-------|
| | | Test | Train | Test | Train | Test | Train |
| $PM_{2.5}$ | out-of-sample | 0.77 | 0.79 | 29.57 | 28.51 | 18.76 | 18.25 |
| | out-of-site | 0.70 | 0.79 | 31.73 | 27.90 | 20.32 | 17.78 |
| | out-of-year | 0.66 | 0.79 | 35.35 | 27.61 | 21.54 | 17.61 |
| $PM_{10}$ | out-of-sample | 0.76 | 0.77 | 51.63 | 50.11 | 35.42 | 34.52 |
| | out-of-site | 0.65 | 0.77 | 57.37 | 49.42 | 39.92 | 33.94 |
| | out-of-year | 0.66 | 0.78 | 60.65 | 49.06 | 40.74 | 33.72 |

### 3.3 Spatial and temporal trends

First, spatial patterns of $PM_{2.5}$ and $PM_{10}$ are analyzed (Fig. S5 and S6). The Indo-Gangetic Plain (IGP) and western arid regions show high levels of $PM_{2.5}$ and $PM_{10}$, especially for years after 2000. Low PM concentrations were observed in south India.

The high terrain in the north and south IGP is unfavorable for pollutant dispersion. Intense human activities in IGP (population > 700 million) emit large amounts of primary PM and gas pollutants ($SO_2$ and nitrogen oxide) coupled with unfavorable dispersion conditions leading to severe PM pollution (Dey et al., 2020; Maheshwarkar et al., 2022). Both $PM_{2.5}$ and $PM_{10}$ concentrations show north-to-south (high-to-low) distribution, consistent with population distribution and corresponding anthropogenic emissions (Upadhyay et al., 2020; Dey et al., 2020).

Figure 2 shows the spatial patterns of seasonal $PM_{2.5}$ and $PM_{10}$ anomalies. The highest PM levels occurred in winter, especially in IGP (positive anomaly > 20 μg m$^{-3}$ relative to the annual mean during 1980-2022). This enhancement is related to additional anthropogenic emissions (from space and water heating of households especially in cold places like IGP ) and stable meteorological conditions (low boundary layer height and low wind speed) (Pandey et al., 2014; Tiwari et al., 2013). During the pre-monsoon (March-April-May), favorable meteorological conditions (increased boundary layer height due to increased temperature and wind speeds) reduce $PM_{2.5}$ concentrations in the IGP area(Dey et al., 2020). During the monsoon season (June to September), rainfall enhances PM deposition, resulting in a substantial reduction of PM concentrations. With the end of the monsoon (post-monsoon, October and November), less rainfall, lower temperatures, extensive open biomass burning (for heating), and reduced boundary layer heights exacerbate PM pollution(Nagpure et al., 2015; Kumari et al., 2021).

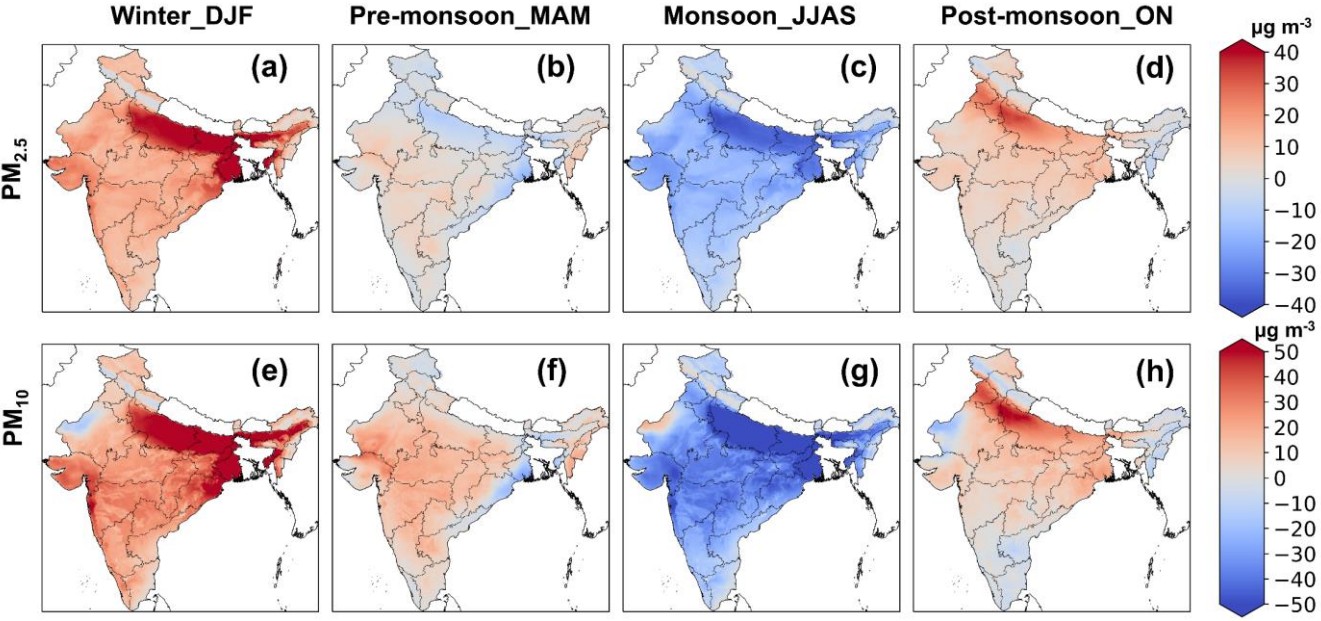

**Figure 2: Spatial patterns of seasonal PM₂.₅ and PM₁₀ anomalies (the difference between seasonal mean and annual mean) in India during 1980-2022.**

The long-term trends of aerosols in India can be better examined given the advantage of long temporal coverage of the LongPMInd dataset. The monthly $PM_{2.5}$ and $PM_{10}$ anomalies from 1980 to 2022 in India and typical regions were first calculated (Fig. 3 and Fig. S7). PM concentrations slowly increased in India (0.19 μg m$^{-3}$ year$^{-1}$) before 2000, the IGP and eastern India increased by 0.43 and 0.26 μg m$^{-3}$ year$^{-1}$, respectively. The PM concentrations jumped in 2000, which can be

attributed to the absence of satellite data for MERRA2 before 2000 (Buchard et al., 2017). The MERRA2 dataset before 2000 could not provide the same level of data quality as in the later period, further leading to a systematic bias in the model estimates. With accelerated industrialization, anthropogenic emissions of primary particulate matter (PPM) and precursors of secondary aerosols (e.g., $SO_2$, NO, and $NH_3$) have increased since 2000 (Pandey et al., 2014; Nagpure et al., 2015), leading to significant increases of PM concentrations in most regions ($p<0.05$), except for western India (Fig. 4 and Fig. S8). $PM_{2.5}$ increased by

0.50 and 0.46 μg m$^{-3}$ per year in the IGP and eastern India during 2000-2017. In early 2018, the Indian government launched the National Clean Air Program (NCAP). The interventions were clubbed into transport, industry, waste management, domestic, and construction activities, road dust, and others (Ganguly et al., 2020). Emissions declined rapidly, and $PM_{2.5}$ concentrations have declined significantly in the IGP (1.63 μg m$^{-3}$ year$^{-1}$), western India (1.22 μg m$^{-3}$ year$^{-1}$), and southern India (0.52 μg m$^{-3}$ year$^{-1}$). However, PM concentrations in east-central India showed an increasing trend (Fig. 4), which may

be related to emissions from mining activities and related industries and thermal power plants (Upadhyay et al., 2020).

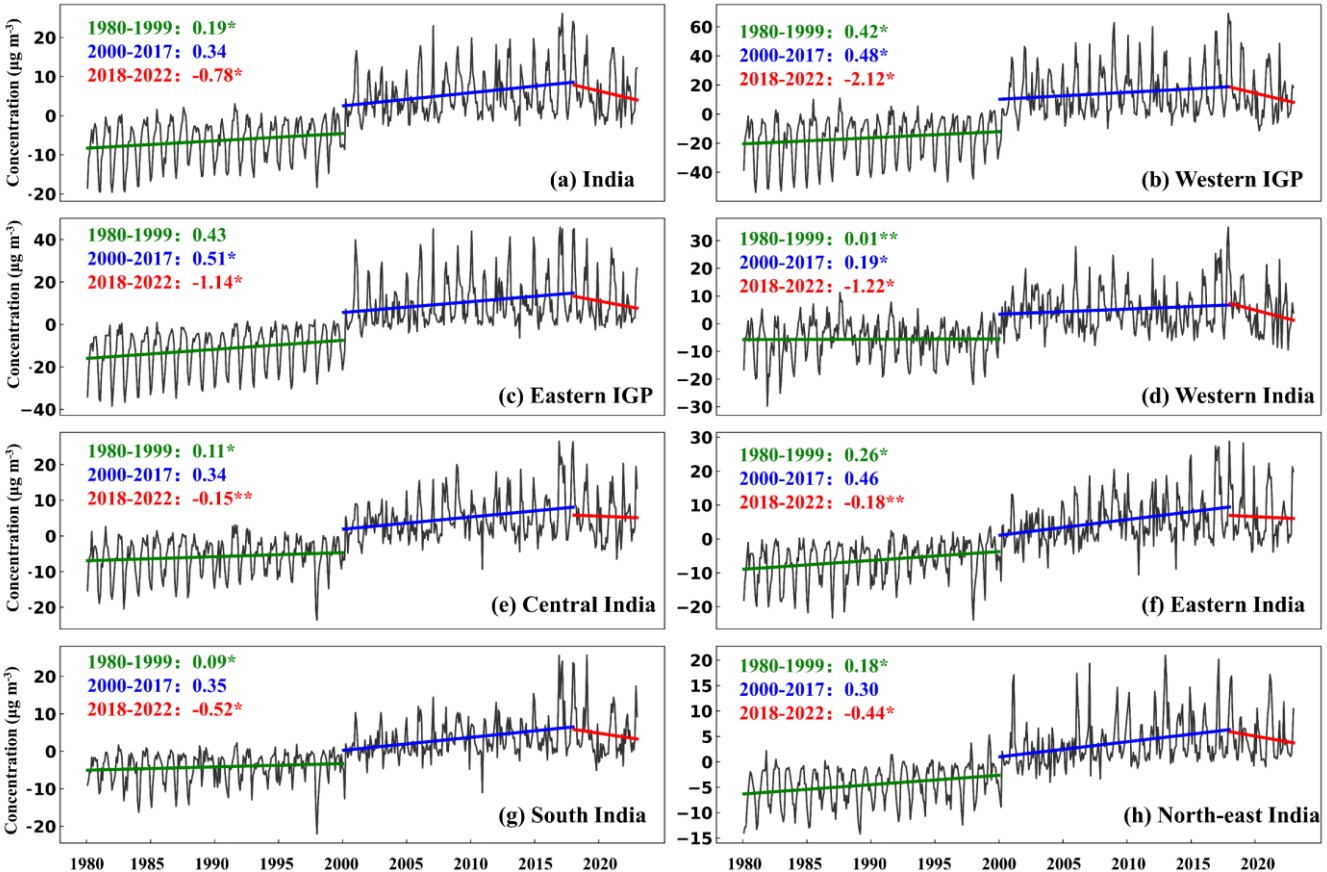

**Figure 3: Time series of monthly $PM_{2.5}$ anomaly from 1980 to 2022 in India and typical regions. The colored straight lines are the linear regression trend (μg m$^{-3}$ year$^{-1}$) for different periods in China, and * represent the significance of the trends (*mean $p < 0.05$ and ** mean $p < 0.01$).**

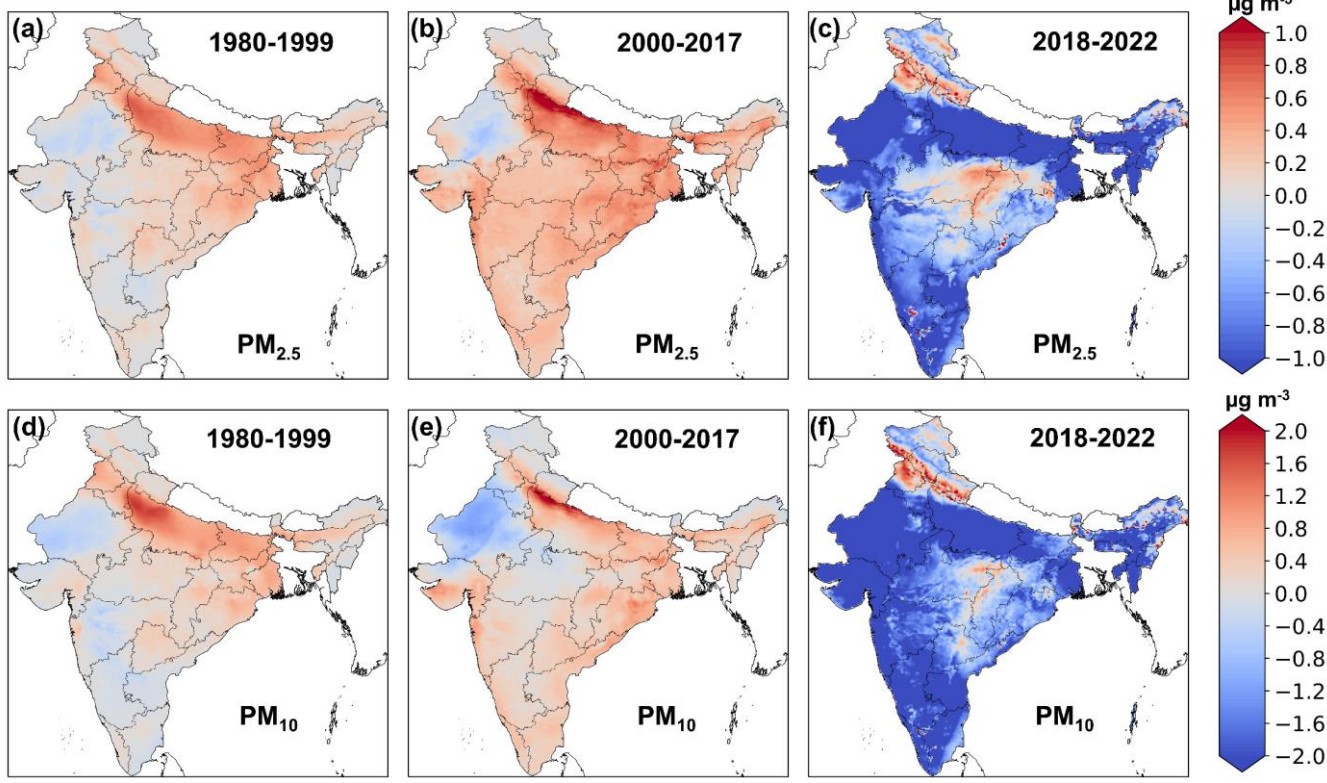

**Figure 4: Spatial patterns of annual changes for PM₂.₅ and PM₁₀ (μg m⁻³ year⁻¹) during different periods (1980-1999, 2000-2017, and 2018-2022).**

### 3.4 Health burden analysis

The health burden of PM$_{2.5}$ was estimated from 2000-2019 following the rapid increase in PM$_{2.5}$ concentrations after 2000.

Using GBD 2019, premature deaths attributed to PM$_{2.5}$ exposure were calculated for six diseases, including ischemic heart disease (CVD_IHD), chronic stroke (CVD_stroke), obstructive pulmonary disease (RESP_COPD), lung cancer (NEO_LUNG), lower respiratory infections (LRI), and diabetes mellitus type 2 (T2_DM) (Murray et al., 2020; Vos et al., 2020).

Figure 5 shows the changes of annual average PM$_{2.5}$ concentrations and corresponding attributed deaths, and Table S2 shows

the uncertainties. PM$_{2.5}$ concentrations showed a fluctuating upward trend with a continuous increase of attributable premature mortality, from 0.73 (95 % Confidence Interval (CI): 0.65-0.80) million in 2000 to 1.22 (95 % CI: 1.03-1.41) million in 2019, with CVD_IHD, CVD_stroke, RESP_COPD, NEO_LUNG, LRI, and T2_DM caused an annual average of 0.35, 0.21, 0.21, 0.02, 0.12, 0.04 million premature mortality, respectively. PM$_{2.5}$-attributable deaths were counted by region (Fig. 5). The IGP had the highest attributable premature deaths, increasing from 0.36 million in 2000 to 0.60 million in 2019, due to high

population density coupled with severe haze pollution (Dey et al., 2020; Pandey et al., 2021).

To reduce premature deaths from PM$_{2.5}$ exposure, policies to mitigate PM$_{2.5}$ pollution should be implemented. In addition, appropriate health advice and enhanced medical facilities to reduce baseline mortality are also important to reduce the health burden (Maji et al., 2023). India has experienced rapid urbanization and large-scale population migration, which introduces uncertainty in health risk estimates for PM$_{2.5}$ (Shi et al., 2020). Country-level baseline disease rates were used, so regional differences were not accounted for due to lack of data, which could introduce some error. In addition, uncertainties in relative risk, population, and PM$_{2.5}$ concentrations may also introduce errors in health risk estimates.

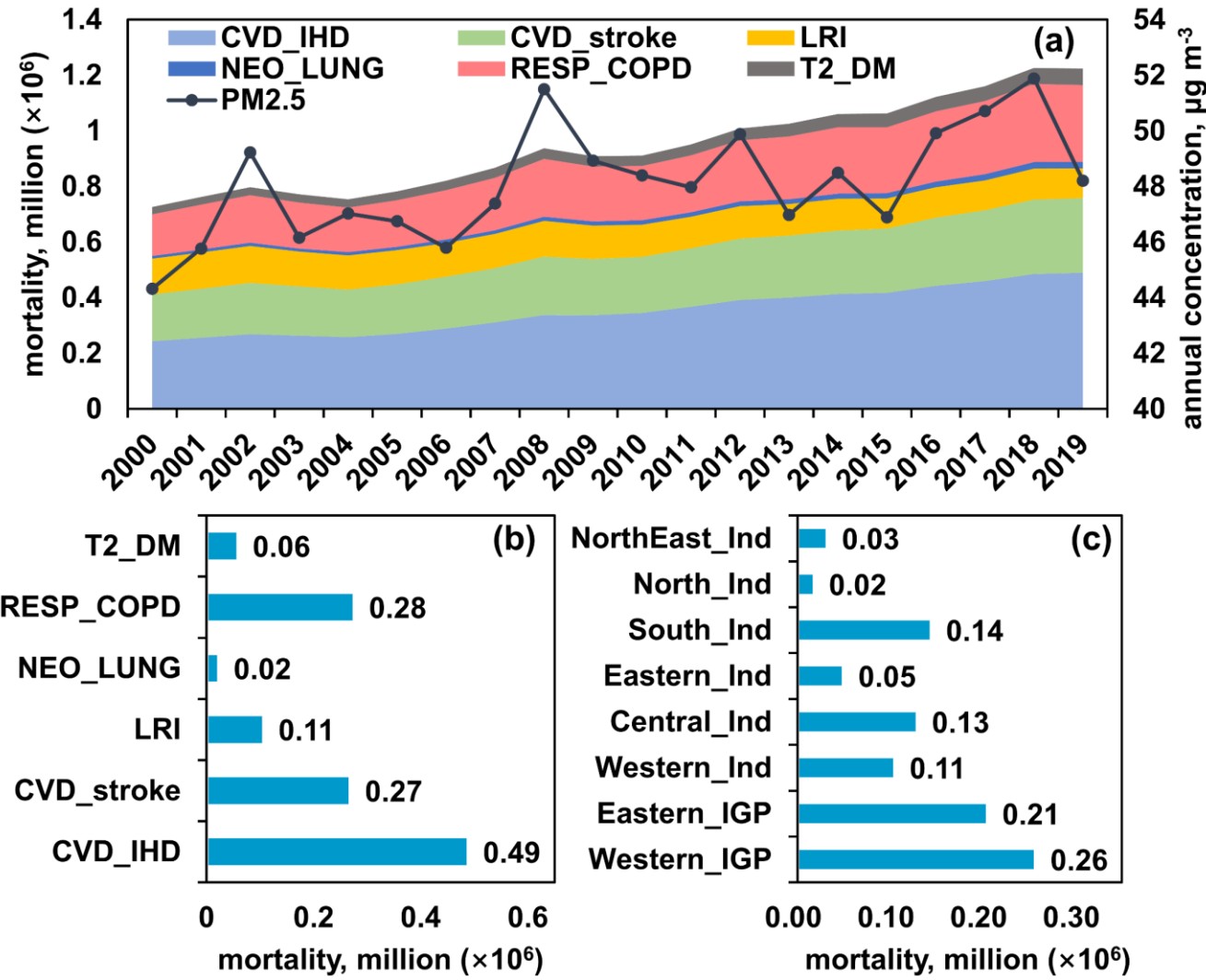

**Figure 5: Annual mortalities due to PM$_{2.5}$-induced diseases in India during 2000-2019, including ischemic heart disease (CVD_IHD), chronic stroke (CVD_stroke), obstructive pulmonary disease (RESP_COPD), lung cancer (NEO_LUNG), lower respiratory infections (LRI), and diabetes mellitus type 2 (T2_DM). Subfigures b and c show statistical results for causes and regions.**

## 3.5 Model complexity

Model complexity can be measured by the number of parameters the model has. As model complexity increases, the model is more capable to learn complex patterns in the data, but at the same time, it may lead to overfitting and inaccurate predictions of new and unseen data (Hu et al., 2021). The impact of the complexity of the tree-based LightGBM model on the performance of training and testing is analyzed. The number of trees (n_estimators) was used as a complexity proxy and the other hyperparameters were kept consistent. All three cross-validation results show that the increase of model complexity improves the model's fitting ability, increasing $R^2$ and decreasing RMSE. However, the increase in complexity did not improve the model's predictive performance. With n_estimators increasing from 100 to 1000, there was no significant change in $R^2$ for the out-of-site and out-of-year CV (-0.01 - 0.01), and the RMSE for the out-of-year CV on the contrary increased by 0.53. Out-of-sample CV showed an improvement in $R^2$ but with limited reduction in RMSE (-2.45). So, using only out-of-sample CV to select hyperparameters and evaluate the model is limiting, and out-of-site and out-of-year CV allows a more objective evaluation of the model's generalization ability.

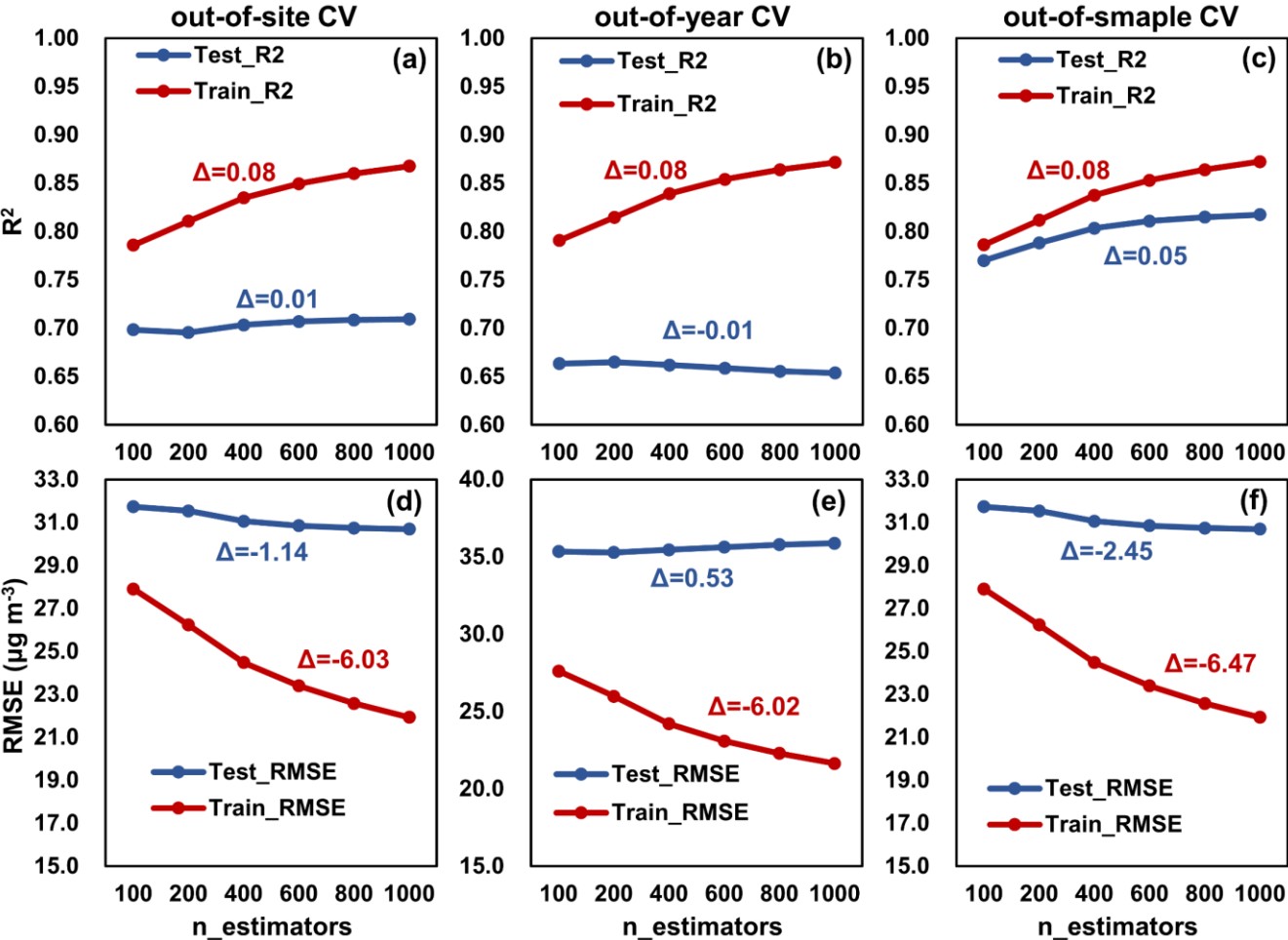

**Figure 6: Three CV results of model complexity test for PM₂.₅ estimation. The n_estimators is the number of trees, representing the complexity of LightGBM. Δ is the difference between the metrics with n_estimators = 1000 and n_estimators = 100. Units of RMSE and MAE are μg m⁻³.**

## 3.6 Uncertainties

Uncertainty in this study comes from two main sources: the machine learning model and the dataset used. Firstly, machine learning is essentially based on probability theory and is influenced by the distribution pattern of the target variable ($PM_{2.5}$ and $PM_{10}$) (Yang et al., 2021b; Breiman, 2001). Due to the low frequency of extreme pollution scenarios, the model suffers from the problem of smoothing predictions, i.e. underestimating high pollution scenarios (Wei et al., 2021a; Yu et al., 2023; Geng et al., 2021). In addition, machine learning has limitations in describing atmospheric physical and chemical processes, and it is difficult to fit complex, logistically long processes, such as secondary aerosol generation (Stirnberg et al., 2021; Li et al., 2023). Attempts have been made to incorporate physical constraints into neural networks to improve interpretability, but this approach is limited to spatially continuous two-dimensional data (Geiss et al., 2022). Other studies have shown that chemical reaction processes can be described by neural networks, but it is still a challenge to efficiently couple them with CTMs (Huang et al., 2022; Huang and Seinfeld, 2022).

The second aspect is the uncertainties caused by the datasets. First, the label (observations) and corresponding features (MERRA2 and ERA5) have a long-tailed distribution with few high pollution records, so there is an issue of imbalance regression (Yang et al., 2021a). The model was trained with a bias towards denser observations, leading to the underestimation of high pollution scenarios. For the problem of imbalanced regression, there are currently main data-based solutions and model-based solutions (Ren et al., 2022a). Data-based solutions require acquiring more data or changing the data distribution by resampling. Model-based solutions increase the weighting of fewer samples (high pollution scenarios) by modifying the loss function. Both methods can improve the accuracy of fewer samples, but they are not suitable for the task of this study because the distribution of the data was altered. Therefore, more observations should be collected in the future to increase observations recorded for high pollution scenarios and mitigate the problem of imbalanced regression. In addition, observational data can only be collected for recent years (2018-2022), which may lead to uncertainties when inference PM concentrations for historical years. In out-of-year validation, the gap between training and testing is mainly attributed to the difference in data distribution among years (data drift, Fig. S2). Besides, changes in climate and human activities over the decades may affect the relationship among emissions, meteorology, and PM concentrations, resulting in extra uncertainty (concept drift).

Secondly, the uncertainty of the input feature sets (ERA5 and MERRA2) also affects the estimation results. The uncertainty of ERA5, a widely used meteorological reanalysis dataset, has been systematically analyzed. ERA5 has good accuracy for most meteorological factors, exceeding other reanalysis data (Muñoz-Sabater et al., 2021; Hersbach et al., 2020). With MODIS data as a reference, the global mean surface temperature of ERA5-Land shows lower uncertainty (Muñoz-Sabater et al., 2021). For precipitation, ERA5 shows 77% correlation with monthly mean Global Precipitation Climatology Project (GPCP) data (Hersbach et al., 2020). Compared to the pre-assimilation data, ERA5-land provides an improved fit to tropospheric winds and humidity (Hersbach et al., 2020).

MERRA2 is a global air pollution reanalysis dataset, published and maintained by NASA, which has been widely used for PM pollution studies in the Indian region, and its reliability has been extensively analyzed (Gueymard and Yang, 2020; Navinya
et al., 2020; Buchard et al., 2017). For MERRA2-AOD, evaluation using AERONET observations showed that MERRA-2 outperformed the Copernicus Atmosphere Monitoring Service (CAMS) in most regions (Gueymard and Yang, 2020). Kumar et al. (2023) predicted ground-level $PM_{2.5}$ concentrations in India using only MERRA2 and machine learning methods, proving the reliability of MERRA2 data.

In addition, before 2000, there was no assimilated satellite data for MERRA-2, and nitrate was not provided in the MERRA2
aerosol reanalysis datasets (Buchard et al., 2017). These issues may be detrimental to the accuracy of the LongPMInd dataset. Previous studies in India have shown that $PM_{2.5}$ estimates based on MERRA2 and empirical formulas suffer from inaccuracies due to issues such as the absence of nitrate, which can be improved by tree-based modeling (Sayeed et al., 2022). The model trained in this study relies heavily on ERA5 (64% relative contribution) with a minor contribution from MERRA2 (36 %). Although tree-based models can improve $PM_{2.5}$ estimation accuracy and the inclusion of ERA5 meteorological features reduces
the model's dependence on MERRA2, model accuracy may decrease for areas dominated by nitrate emissions and for years before 2000.

**Data availability**

The LongPMInd dataset, including daily $PM_{2.5}$ and $PM_{10}$ concentration (10km) for India during 1980-2022 is publicly accessible. All data are provided in NetCDF format and can be downloaded at https://zenodo.org/records/10073944 (Wang et
al., 2023a).

**Supporting Information**

Research domain, feature importance, spatial and temporal patterns of $PM_{2.5}$ and $PM_{10}$, and uncertainty of estimated annual mortalities.

**Author contribution**

**Shuai Wang**: Methodology, Software, Writing - original draft. **Mengyuan Zhang**: Visualization, Validation. **Hui Zhao**: Data curation, Methodology. **Peng Wang**: Methodology, Writing - reviewing and editing. **Sri Harsha Kota**: Data curation. **Qingyan Fu**: Writing - reviewing and editing. **Cong Liu**: reviewing and editing. **Hongliang Zhang**: Conceptualization, Funding acquisition, Supervision, Writing - reviewing and editing.

**Competing interests**

The contact author has declared that neither they nor their co-authors have any competing interests.

**Acknowledgment**

This work was supported by the National Key R&D Program of China (2022YFC3701105), Co-fund DFG-NSFC Sino-German AirChanges project (448720203), National Natural Science Foundation of China (42077194/42061134008), and Shanghai International Science and Technology Partnership Project (No. 21230780200).

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
