# Peer review of "Reconstructing long-term (1980-2022) daily ground particulate matter concentrations in India (LongPMInd)"

_Earth System Science Data, 2024_

## Author Response (AR1)

**Point-by-point responses to the comments & suggestions from the editor and reviewers**

**Journal: Earth System Science Data**

**Manuscript ID: essd-2024-34**

**Title: "Reconstructing long-term (1980-2022) daily ground particulate matter concentrations in India (LongPMInd)"**

**Comments from Reviewer #1:**

Based on the Light Gradient Boosting Machine (LightGBM), this paper constructs a model for fusing multi-source data and estimating the long-term (1980-2022) historical daily ground PM datasets in India (LongPMInd). This study supplements data for regions in India lacking observation sites based on available data, providing data support for future research in areas such as air quality, public health, and climate. In light of these considerations, the manuscript could be suitable for publication after addressing the following minor comments.

- We truly appreciate the time and effort you have devoted to carefully evaluating our submission and providing us with valuable suggestions for improvement. In the revision, we carefully revised the manuscript based on these comments.

Comments:

L41-51: There are various methods for estimating ground PM5. Based on the analysis of the data obtained from these methods, are there differences in the PM2.5 in India?

- Thanks for the comment. Two main approaches have been used for large-scale and long-term $PM_{2.5}$ estimation: scaling methods and statistical methods. Scaling methods use chemical transport modeling (CTM) to simulate the association between aerosol optical depth (AOD) and

PM$_2$, which requires no ground observations. However, the relationship between PM$_{2.5}$ -AOD is spatially and temporally variable and without the constraints of ground observations, this method usually has a large uncertainty (Ma et al., 2022). Compared with scaling methods, statistical methods based on multivariate data fusion have higher prediction accuracy and have been widely used. Statistical models (traditional linear and nonlinear regression and machine learning algorithms) estimate PM$_{2.5}$ concentrations by fitting the relationship between input variables (meteorological, emissions, and other relevant variables) and target variables (Wang et al., 2023b; Wei et al., 2021a; Ren et al., 2022; Katoch et al., 2023).

The PM$_{2.5}$ concentrations in India obtained by different methods differ in terms of accuracy, spatial and temporal coverage, and spatial and temporal resolution. Accuracy is the most discussed metric. Tree-based machine learning (ML) models typically outperform deep learning approaches and traditional machine learning methods such as Lasso, KNN, and SVM in tabular data (e.g., air pollutant observation datasets), and thus have been widely developed and used (Grinsztajn et al., 2022; Sayeed et al., 2022; Wei et al., 2023). Due to the serious non-random missing of AOD satellite data, the estimated PM$_{2.5}$ is spatially and temporally discontinuous. Bai et al. (2024) and Wei et al. (2023) filled in the missing AOD based on the tensor flow AOD reconstruction algorithm and 4D-spatio-temporal extreme tree algorithm, respectively, to realize the spatial and temporal full-coverage prediction of PM$_{2.5}$. The spatial resolution of the estimated PM$_{2.5}$ concentration depends on the resolution of the input data, e.g. ERA5-land provides a product with a resolution of 0.1 ° x 0.1 ° and MERRA2 provides a product with a resolution of 0.5 ° x 0.625 °. MODIS satellites can provide products with a resolution of up to 1 km.

**Changes in Lines 47-54:** "Two main approaches have been used for large-scale and long-term PM$_{2.5}$ estimation: scaling methods and statistical methods. Scaling methods use chemical

transport modeling (CTM) to simulate the association between aerosol optical depth (AOD) and $PM_{2.}$, which requires no ground observations. However, the relationship between $PM_{2.5}$ -AOD is spatially and temporally variable and without the constraints of ground observations, this method usually has a large uncertainty (Ma et al., 2022). Compared with scaling methods, statistical methods based on multivariate data fusion have higher prediction accuracy and have been widely used. Statistical models (traditional linear and nonlinear regression and machine learning algorithms) estimate $PM_{2.5}$ concentrations by fitting the relationship between input variables (meteorological, emissions, and other relevant variables) and target variables (Wang et al., 2023b; Wei et al., 2021a; Ren et al., 2022; Katoch et al., 2023)."

References need to be carefully checked:

L56-58: References should be cited in the following format: "Wei et al. (2021)".

    - Thanks for the comment. We have double-checked and revised the formatting of the references.

L216: Subscript.

    - Thanks for the comment. We apologize for our carelessness, we have revised and double-checked the formats.

L298-299: Format of the references.

    - Thanks for the comment. We have changed the format of the references and double-checked the manuscript.

There are some missing parts in the References, such as L396 and 435.

- Thanks for the comment. We have double-checked and added the missing parts of the references.

L62-63: What exactly is the insufficient model robustness and implementation capacity? Is it caused by a lack of training data or is it a flaw in the model itself?

- Thanks for the comment. the insufficient model robustness and implementation capacity include both training data and model evaluation. On the one hand, observation data are lacking and observation sites are unevenly distributed, which is not conducive to model training. On the other hand, the cross-validation accuracy of the model does not reflect the accuracy of the predictions for unobserved areas and years, due to the differences of data distribution between different regions and years. We have added the corresponding statement in the Introduction.

**Changes in Lines 77-79:** "However, it is challenging to establish long-term, full-coverage, high accuracy, open-source PM data products locally in India due to insufficient observational data and lack of model robustness due to variations of data distribution across regions and years"

The article mentions many machine learning methods, but the choice of the LightGBM method was rather abrupt, with features such as simple structure, high efficiency and robustness not supported by data and literature.

- Thanks for the comment. The Light Gradient Boosting Machine (LightGBM) is an optimized Gradient Boosting Decision Tree (GBDT). It uses Histogram's decision tree algorithm along with Gradient-based One-Side Sampling (GOSS), which can save memory and computation time (Ke et al., 2017). Our previous study comparing several commonly used machine learning models showed that the Light Gradient Boosting Machine (LightGBM) has similar performance to

the eXtreme Gradient Boosting (XGBoost) model with the highest accuracy, but LightGBM was faster and more robust and therefore has the potential to estimate long-term concentrations of PM in India. We have added the corresponding statement in the Introduction.

**Changes in Lines 83-89:** "The Light Gradient Boosting Machine (LightGBM) is an optimized Gradient Boosting Decision Tree (GBDT) (Ke et al., 2017), and has shown superior performance in many fields (Wei et al., 2021b; Yan et al., 2021; Sun et al., 2020; Liang et al., 2020). LightGBM uses Histogram's decision tree algorithm along with Gradient-based One-Side Sampling (GOSS), which can save memory and computation time (Ke et al., 2017). Our previous study comparing several commonly used machine learning models and found that the LightGBM has similar performance to the eXtreme Gradient Boosting (XGBoost) with the highest accuracy, but LightGBM was faster and more robust, which has the potential to estimate long-term concentrations of PM in India (Wang et al., 2023a)."

L79: Why data larger than 99.99% should be excluded.

- Thanks for the comment. Extreme values can affect the stability of the model, leading to poorer generalization to unseen data, and by trimming these extreme values, the model can be more robust. In addition, extreme values may lead to a very uneven distribution of the data, making the model more inclined to accommodate extreme values while ignoring most of the normal conditions in the dataset, leading to a decrease in model performance.

In our dataset, the total sample size is about 230,000. the 99% quantile is 483, and there are only 23 trees larger than this value, so we think it is reasonable to exclude this value.

**Changes in Lines 99-100:** "As extreme values affect model robustness, Observations data less than 0.01 % and larger than 99.99 % were excluded."

Is there any standard for the choice of meteorological factors. For example, why was evaporation considered, and why was humidity not used directly. It was mentioned in L92 that features were filtered with relative importance, so which features were used for selection before and which parameters were filtered out in this step?

- Thanks for the comment. Feature selection was performed using relative importance, which was calculated using the Gain. Features with relative importance less than 0.02 were excluded. Humidity was not used because ERA5-land does not provide humidity data, we obtained a lower resolution (0.25°) humidity data from ERA5 on pressure levels, but it had a lower importance and was therefore not included. In addition, the relative humidity can be calculated by the temperature and dewpoint temperature, so we think it is reasonable to exclude the relative humidity. Our results show that evaporation has a relatively high importance (7%) and is therefore included. Two meteorological variables, relative humidity, and surface albedo, were filtered through feature importance in this step.

**Changes in Lines 100-102:** "The feature was selected by the relative importance, which was calculated by the Gain, and several meteorological factors with high relative importance are included (Table 1)."

Figure 1: The panel below should be (b) PM10.

Figure 2: Missing labels in the second column.

- Thanks for the comment. We apologize for our carelessness, we modified Figures 1 and 2 and examined the manuscript carefully.

Table S2: Column names are also capitalized to match the content of the article.

- Thanks for the comment. We've changed the column names to capitalization to match the content of the article.

Figure 5: Numbers in (b) should be kept to two decimal places and axes are adjusted according to the range of CVD_IHD to ensure a complete presentation of the data.

- Thanks for the comment. We've changed the numbers to two decimal places and adjusted the axes according to the data range in Figure 5.

L109: The concept of GBD is appearing for the first time and should be labeled with its full name.

- Thanks for the comment. We have added the full name of GBD, which is " Global Burden of Disease".

**Changes in Lines 136:** "According to the database of Global Burden of Disease Study (GBD) in 2019"

L119: Does it mean that people will experience health effects related to PM2.5 when the concentration of PM2.5 is in this range?

- Thanks for the comment. We apologize for the lack of clarity, but this represents the minimum exposure level for $PM_{2.5}$ health risk, below which there is no health risk considered.

**Changes in Lines 148-149:** "The minimum-risk exposure level for the health effects of $PM_{2.5}$ are in the range of 2.4 to 5.9 $\mu g\ m^{-3}$."

L140: "RMSE (35.35 and 60.65 μg m-3) and MAE(21.54 and 40.74 μg m-3)" I don't think they can be described as small.

- Thanks for the comment. We modified the description. The model's prediction accuracy for unobserved years decreases slightly compared to out-of-sample CV ($R^2$ decreases by 14% and RMSE increases by 20%) due to differences in data distribution among years.

**Changes in Lines 170-172:** "The model's prediction accuracy for unobserved years decreases slightly compared to out-of-sample CV ($R^2$ decreases by 14% and RMSE increases by 20%) due to differences in data distribution among years (Fig. S2)."

L241: PM"1"? Was it a mistake?

- Thanks for the comment. We apologize it was a mistake. Here it should be $PM_{2.5}$ and $PM_{10}$.

**Changes in Lines 273-275:** "Firstly, machine learning is essentially based on probability theory and is influenced by the distribution pattern of the target variable ($PM_{2.5}$ and $PM_{10}$)"

The references in the introduction are can be reinforced. For example, the random forest and LightGBM are also used to construct PM2.5 and ozone data in China (e.g., Li et al., 2021; Ni et al., 2024).

Table 1: it seems that the authors mainly used aerosol diagnostics from MERRA-2 as the proxy variables to derive long-term PM dataset in India. Since no nitrates were provided in the MERRA-2 aerosol reanalysis, the authors should discuss the potential impacts on the modeling accuracy in the manuscript.

- Thanks for the comment. Indeed, nitrate was not provided in the MERRA-2 aerosol reanalysis datasets. Previous studies in India have shown that $PM_{2.5}$ estimates based on MERRA2 and empirical formulas suffer from inaccuracies due to issues such as the absence of nitrate, which can be improved by tree-based modeling. In addition, the model trained in this study relies heavily on ERA5 (64% relative contribution) with a small contribution from MERRA2 (36%). We must recognize that although tree-based models can improve $PM_{2.5}$ estimation accuracy and the inclusion of ERA5 meteorological features reduces the model's dependence on MERRA2, model accuracy may decrease for areas dominated by nitrate emissions.

**Changes in Lines 309-316:** "In addition, before 2000, there was no assimilated satellite data for MERRA-2, and nitrate was not provided in the MERRA2 aerosol reanalysis datasets (Buchard et al., 2017). These issues may be detrimental to the accuracy of the LongPMInd dataset. Previous studies in India have shown that $PM_{2.5}$ estimates based on MERRA2 and empirical formulas suffer from inaccuracies due to issues such as the absence of nitrate, which can be improved by tree-based modeling (Sayeed et al., 2022). The model trained in this study relies heavily on ERA5 (64% relative contribution) with a minor contribution from MERRA2 (36 %). Although tree-based models can improve $PM_{2.5}$ estimation accuracy and the inclusion of ERA5 meteorological features reduces the model's dependence on MERRA2, model accuracy may decrease for areas dominated by nitrate emissions and for years before 2000."

Line 94: "The meteorological and emission features contributed 64% and 31% to the PM2.5 prediction." How should we interpret this result? Why did the emission play a much weak role than meteorological conditions in PM2.5 prediciton? Also, what is the reason for the largest importance of SP?

- Thanks for the comment. Relative importance is a metric used to measure the influence of features on the model's predictions. It reflects the contribution of each feature to the model's predictions. Features with higher relative importance have a greater influence on the model's predictions.

Meteorology is more important than emissions for two main reasons:

1. Data quality. Compared to MERRA5, which has higher uncertainty and lower spatial resolution, ERA5 has higher resolution and accuracy, and the meteorological features can provide richer information and contribute more to model training, thus having higher importance.

2, Number of features. The number of meteorological features is 10, which is more than the emission-related features (6). The average importance of each meteorological feature is 6.5% and the average importance of each emission feature is 5.2%, which is not a big difference. Thus the larger number of meteorological features had a higher cumulative relative importance.

Surface pressure shows the highest importance. Firstly, surface pressure has an important effect on $PM_{2.5}$ concentration. On the one hand, high-pressure systems can lead to stagnant atmospheric conditions, which are not conducive to $PM_{2.5}$ dispersion. On the other hand, atmospheric pressure can indirectly influence $PM_{2.5}$ concentration by influencing other meteorological factors. For example, low-pressure systems accompanied by high humidity could

affect PM$_{2.5}$ nucleation, condensation, and coagulation, further leading to higher PM$_{2.5}$

concentrations.

Secondly, the high quality of surface pressure data provides richer information, which in turn

shows high relative importance. Numerical models can accurately simulate surface pressure

because it is largely controlled by large-scale atmospheric motions, which are well simulated by

these numerical models and can be further improved through advanced data assimilation

techniques. Challenges remain in the simulation of some meteorological factors, such as wind

speed, which is subject to uncertainty at smaller local scales due to simplified parameterization

and local surface heterogeneity, etc.

**Changes in Lines 116-121:** "Meteorology is more important than emissions. Compared to

MERRA5, which has higher uncertainty and lower spatial resolution, ERA5 has higher resolution

and accuracy, and the meteorological features can provide richer information and contribute more

to model training, thus having higher importance (Muñoz-Sabater et al., 2021; Hersbach et al.,

2020). Besides, more numbers of meteorological features were used to train the model, thus

contributing more to prediction results. The highest importance of surface pressure can be

attributed to the important effect to PM$_{2.5}$ concentration and its high data quality (Chen et al.,

2020; Bauer et al., 2015)."

Section 2.3: the equation should be numbered. How was RR calcualted? The authors provide no

descriptions about this important factor.

- Thanks for the comment. Thanks, we have added numbers to the equations. The relative risk

is a discrete value obtained from GDB2019, which is the relative risk corresponding to each PM$_{2.5}$

concentration level. GDB 2019 calculates the aggregated relative risk by quantitative systematic overview and estimates $PM_{2.5}$ concentration data from various sources such as ground observation, remote sensing, etc., and then synthesizes the relative risk under different $PM_{2.5}$ exposure levels. We have added the appropriate content in Section 2.3.

**Changes in Lines 145-147:** "The relative risk is a discrete value obtained from GDB 2019, which is the relative risk corresponding to each $PM_{2.5}$ concentration level. Details about the calculation way of relative risk can be found in GBD 2019 (Murray et al., 2020)."

Table 3: as suggested by the results, the prediction model appeared to suffer from signficant overfitting, in particular at the year scale. What are possible reasons?

- Thanks for the comment. The gap between training and testing performance is caused by several reasons, including overfitting as well as data and concept drift. In out-of-sample cross-validation, the training and testing sets were randomly divided, so their distributions are the same, and the gap between training and testing should be attributed to model overfitting. In this study, there is a small gap between training and testing in the out-of-sample cross-validation (delta RMSE is 4 % for $PM_{2.5}$). So the model showed a low risk of overfitting. However, the data distributions were different among years (Fig. S2). In out-of-year validation, the gap between training and testing is mainly attributed to the difference in data distribution (data drift) and is therefore not an overfitting problem. In addition, the drivers of $PM_{2.5}$ pollution in India may have changed considerably over decades (e.g., changes in the main emission sources), thus resulting in a decrease in the accuracy of models trained with data from recent years ( concept drift). We have added this section to the discussion.

**Changes in Lines 293-295:** "In out-of-year validation, the gap between training and testing is mainly attributed to the difference in data distribution among years (data drift). Besides, changes in climate and human activities over the decades may affect the relationship among emissions, meteorology, and PM concentrations, resulting in extra uncertainty (concept drift)."

Figure 3: the trend should be estimated using piece-wise linear regression. Also, what are possible reasons for an abrupt jump in PM concentrations in 2000 and 2015?

- Thanks for the comment. Thanks, we have performed piecewise linear regression. The data were split into three phases 1980-2000, 2000-2018, and 2018-2022 (Figure 3). The PM$_{2.5}$ concentrations jump in 2000 can be attributed to the absence of satellite data for MERRA2 before 2000. The MERRA2 dataset before 2000 could not provide the same level of data quality as in the later period, further leading to a systematic bias in the model estimates. There was no significant jump of concentration in 2015, are you referring to the turning point in 2018? Here we attributed it to the launch of the National Clean Air Program by the Government of India at that time. The interventions were clubbed into transport, industry, waste management, domestic, and construction activities, road dust, and others. Emissions declined rapidly, and PM$_{2.5}$ concentrations have shifted from an increasing to a decreasing trend in most areas (Figure 4). We have added the corresponding discussion.

**Changes in Lines 214-216:** "The PM concentrations jumped in 2000, which can be attributed to the absence of satellite data for MERRA2 before 2000 (Buchard et al., 2017). The MERRA2 dataset before 2000 could not provide the same level of data quality as in the later period, further leading to a systematic bias in the model estimates."

**Changes in Lines 221-224:** "The interventions were clubbed into transport, industry, waste management, domestic, construction activities, road dust, and others (Ganguly et al., 2020). Emissions declined rapidly, and $PM_{2.5}$ concentrations have declined significantly in the IGP (1.63 $\mu g\ m^{-3}\ year^{-1}$), western India (1.22 $\mu g\ m^{-3}\ year^{-1}$), and southern India (0.52 $\mu g\ m^{-3}\ year^{-1}$)."

**Reference**

[revised manuscript text omitted]

---

## Author Response (AR2)

**Point-by-point responses to the comments & suggestions from the editor and reviewers**

**Journal: Earth System Science Data**

**Manuscript ID: essd-2024-34**

**Title: "Reconstructing long-term (1980-2022) daily ground particulate matter concentrations in India (LongPMInd)"**

- We truly appreciate the time and effort of reviewers and editor have devoted to carefully evaluating our submission and providing us with valuable suggestions for improvement. We have checked the manuscript carefully again to make sure there were no mistakes. Once again, we would like to thank the editor and reviewers and other staff for their dedication.